# Investment Evaluation and Partnership Selection Model in the Offshore Wind Power Underwater Foundations Industry

**Min-Yuan Cheng and Yung-Fu Wu \***

Department of Civil and Construction Engineering, National Taiwan University of Science and Technology, Taipei 10672, Taiwan; myc@mail.ntust.edu.tw
\* Correspondence: d10405008@mail.ntust.edu.tw; Tel.: +886-2-2733-0004

**Abstract:** With a plan to achieve a target of 5.7 GW offshore wind power capacity in 2025, Taiwan anticipates building a 36-billion USD industry, which makes Taiwan a center of attention in the global marketplace of civil engineering construction. Aimed at Taiwan's underwater foundations industries, this study is the first to develop an investment evaluation model (IEM) by applying FPR to obtain risk factor weights and calculate the overall investment risk value with a numerical scoring method. In a context where no precedent exists for reference, this study provides auxiliary and supportive tools to help builders to make the decision, based on objective indicators, whether to undertake an investment. To date, no research has been conducted to introduce a reasonable mathematical model that discusses the issue of partner selection in the field of offshore wind power. This study is the first paper to construct a SWARA-FTOPSIS partner selection model, which enables underwater foundations builders to take specific Taiwanese characteristics into account in their selection of the best partners to meet transportation, construction, and installation requirements. Finally, the study uses the case of the Taipower Offshore Wind Power Project (2nd phase) to verify the feasibility of this model.

**Keywords:** offshore wind; underwater foundations; multi criteria decision making (MCDM); partnership selection; preference relation theory (PRT); step-wise weight assessment ratio analysis (SWARA); fuzzy technique for order preference by similarity to ideal solution (FTOPSIS)





## 1. Introduction

For many developing countries, access to new types of energy may be essential for economic development. Recent studies have shown a direct relationship between the level of development of a country and its energy consumption regime. Given the limited reserves of fossil fuels and the rising level of energy demand across the world, it is no longer sustainable to rely solely on existing energy sources. Fortunately, many countries have realized the benefits of taking advantage of various energy sources, especially renewable sources, in meeting their current and future energy needs, and are heavily investing in the research and development of the technologies and industries that are needed to tap into these essentially unlimited sources of energy [1]. Wind power is one of the cleanest energy sources. The pollution produced in the process of wind power generation is significantly lower than that of other green energies; it can come close to nearly "zero carbon emission." Therefore, wind power is becoming the most valued power industry amidst the hype of green energy development in the 21st century [2]. According to World Energy Outlook 2019, published by the International Energy Agency, offshore wind power has the technological potential to satisfy several times the current electricity demand and, within 10 years, will have a cost comparable to fossil fuels. It is projected that the installed capacity will jump from 23 GW in 2018 to 345 GW in 2040 and the power generation will increase 15 times, accounting for 13% of total global power generation. Backed by the government's policies and the goal of reduced cost of technology, the industry is expected to grow into a 1-trillion USD industry in the next 20 years [3].

According to the Taiwanese government's plan, the offshore wind power industry will reach an autonomous supply with a cumulative installation of 5.7 GW in 2025 with the promotion of energy diversification. It is estimated that clean electricity will reach 21.5 billion kWh each year to create an approximately 36-billion USD cumulative investment and 20,000 jobs [4]. At present, "Block A" is being planned assiduously with an annual increase of 1 GW in an effort to advance the promotion of offshore wind energy. The plan includes an additional post-2025 annual increase of 1 GW of offshore wind power installation capacity in Taiwan's waters. By 2030, an additional 5 GW of installation capacity is expected [5]. Taiwan's offshore wind farms project is considered one of the largest and most targeted construction sites in the world and is therefore incredibly attractive to top offshore wind power developers.

Offshore wind power construction can be roughly categorized into seven major areas: wind farm development, wind turbine systems, wind turbine component manufacturing, underwater infrastructure, maritime engineering, wind farm operation and maintenance, and green finance [6]. Among them, the most crucial part pertaining to the success of the construction is that the underwater foundation must withstand the test of extreme weather, complex terrain, and firmly support wind power equipment to ensure that the wind turbine can operate normally for at least 20 years at sea. The commercial opportunities in offshore wind power development are highly desirable; however, the construction requires huge capital and the return on investment takes an extended period of time in the midst of other unknown challenges and risks [7]. Moreover, energy expertise, as well as complex external factors such as policy variability, emerging technologies, and the increasingly volatile international energy market [8], must also be taken into consideration. Local Taiwanese manufacturers often have no basis for judgement and thus find it difficult to decide whether to partake in the industrial supply chain. It is therefore particularly important that they be well equipped with appropriate decision-making methods in order to formulate a reasonable investment plan. In addition, while underwater foundation procurement strategies include foundation pile manufacturing, ship transportation, installation, construction, etc., few companies in the existing supply chain of the offshore wind power industry are able to undertake all project items individually. Currently in the market, the underwater foundations manufacturer works as the main contractor, who makes the decision whether to make an investment in the offshore wind power industry. Hence, even if all the builders are equipped with the right transportation and installation capabilities to co-participate in market bidding, the selection of the most suitable partner can still pose a challenge to be resolved.

This study explores the decision-making of offshore wind power underwater infrastructure investment evaluation and partner selection, which makes it the first research to construct a two-stage decision-making model. The first stage is the Investment Evaluation Model, to allow domestic companies to pre-evaluate the investment risks in the offshore wind power underwater infrastructure industries. The second stage is the Partner Selection Model, to select the most suitable partners to participate in the bidding so as to maximize the chance of a successful cooperation and future operating profits. Details are as follows:

1. Investment Evaluation Model (IEM):

Through literature review, it was found that the multiple criteria decision making (MCDM) method is widely used in renewable energy investment decision making; its effectiveness is thus self-evident. Among the methods, the analytic hierarchy process (AHP) method collects the opinions of scholars and experts and employs a concise hierarchical structure to systematize complex assessment questions. Pairwise comparisons of factors at each level are carried out through nominal scales and the quantified values are used as a reference to assess acceptance, which is the most commonly used method in performance evaluation in renewable energy projects. Karatop et al. (2021) used fuzzy AHP to determine how to make the best investment decision in Turkey's renewable energy sector [9]. Wu et al. (2019) used fuzzy AHP to evaluate portfolio selection of renewable energy investment projects under different corporate strategic scenarios [10]. Karakaş et al. (2019) modified

fuzzy AHP to determine the most suitable renewable energy alternative for Turkey and formulate a reasonable energy investment plan [11]. Nigim et al. (2004) used AHP for a preliminary renewable energy-related feasibility study to assist in evaluating the most suitable development projects [12]. Keeley et al. (2018) used AHP to analyze the relative importance of determinants of wind and solar energy for foreign investment in developing countries [13]. Ahmad et al. (2014) used AHP to select renewable energy sources for the sustainable development of Malaysia's power generation system [14]. Aragonés Beltrán et al. (2014) used AHP to help Spanish renewable energy developers decide whether to invest in a specific solar thermal power plant project [15]. Chatzimouratidis et al. (2009) used AHP to evaluate the technology, economy, and sustainability of power plants [16]. Lee et al. (2009) used AHP to analyze multiple factors that affect the success of wind farm operations and help select suitable wind farm projects [17]. Nevertheless, AHP still has its shortcomings. (1) When the numbers of criteria, levels, or evaluation attributes increase, the pairwise comparison will become quite complicated, resulting in lower efficiency. (2) When there is uncertainty or incomplete information in the standard attributes, it becomes difficult to compare the two. (3) The larger the number of candidate solutions or criteria is, the more questions there are to be answered, making it more difficult to meet the consistency index requirements [18–20]. The fuzzy preference relations (FPR) proposed by Herrera-Viedma et al. (2004) not only inherited the characteristics and advantages of AHP, but also improved the inconsistency caused by multiple decision makers, multiple criteria, or multiple candidate solutions. By reducing the number of pairwise comparisons, it makes the calculation simpler and easier, which improves efficiency and accuracy [21].

Aiming at Taiwan's underwater foundations industries, this innovative research develops the Investment Evaluation Model (IEM), applies FPR to obtain the risk factor weights, and calculates the overall investment risk value with a numerical scoring method. In a dilemma where no precedents exist for reference, the study provides auxiliary and supportive tools to help manufacturers make a decision, based on objective indicators, whether to undertake the investment.

2.　　Partner Selection Model (PSM):

From a strategic perspective, different organizations form alliances (a form of partnership) to jointly invest in emerging industries; they overcome local market restrictions, diversify risks, and gain mutual benefits [22]. Past studies have proven that partner selection through logical and scientific methods increases the rate of success in alliances [23]. Determine the appropriate methods to support decision making is a vital step in the partner selection process [24]. Past papers have introduced a whole host of methods such as an improved version of the genetic algorithm (GA) [25], intuitionistic fuzzy set [26], grey fuzzy valuation method based on entropy method and analytic hierarchy process (AHP) [27], particle swarm optimization (PSO), improved technique for order of preference by similarity to ideal solution (TOPSIS) [28], the cooperative partner selection model based on fuzzy evaluation, and improved TOPSIS [29]. However, the majority of the existing decision-making research contains information distortions and weight deviations and at the same time lacks consideration for the relationship and mutual relevance between the evaluation indicators. In the selection process, decision makers are often subjective, ambiguous, and unable to give an objective evaluation based on accurate data. As a result, the accuracy of the evaluation is affected to a certain extent [30].

To date, no research has been conducted to introduce a reasonable mathematical model that discusses the issue of partner selection in the field of offshore wind power. This study has defined partnership as: in order to meet client expectations and accomplish client goals, two or more companies willingly forgo their own independent business models to build an alliance (a form of partnership). With respective core competences and competitive advantages, they enter emerging markets and undertake risks and create value. They support each other to achieve unprecedented performance and organically develop a mutually beneficial long-term cooperative relationship. Based on this viewpoint, this study is the first paper to construct a SWARA-FTOPSIS partner selection model.

From the perspective of offshore wind power underwater foundations builders, the study analyzes the impact factors of the constructors' partnership and adopts a step-wise weight assessment ratio analysis (SWARW) to calculate the impact factor index weights. SWARA was proposed by Keršuliene et al. in 2010. With a clear and easy-to-understand logic, it is an uncomplicated method that is easy to use [31]. Its key feature is to invite knowledgeable experts to individually judge the importance of each standard and then combine different judgments or priority levels with the geometric mean method to obtain the overall result. Such a negotiation improves the accuracy of the calculation method. It takes the mutual influence and correlation of various factors into account, which avoids the shortfalls of other weighting methods [32–34]. Finally, the fuzzy technique for order preference by similarity to ideal solution (FTOPSIS) is adopted to come up with the ranking of the candidates. Fuzzy TOPSIS was proposed by Chen (2000), which combines the TOPSIS method with the fuzzy logic theory. It is aimed at solving the problem of linguistic ambiguity and the need for collective decision-making. It not only eliminates the ambiguity caused by human judgment in the decision-making process, but also the bias of the decision makers who, due to their personal preference for the best of standards, neglect to consider the most appropriate solution between the best and the worst scenarios [35,36]. This paper proposes an innovative combination of the SWARA-FTOPSIS method, which effectively enhances the scientificity and accuracy of decision analysis. It also improves on the omissions caused by imprecise evaluation values and blurred human judgments. This research works to help managers make better decisions in different circumstances.

## 2. Literature Review

### 2.1. Risk Factors of Investment in Offshore Wind Power Industry

This study compiles a total of 12 articles on industrial investment risks from domestic and foreign literatures such as renewable energy, onshore and offshore wind farms, marine engineering, and transnational cooperative construction. 26 factors influencing offshore wind power investment have been analyzed, as elaborated in Table 1.

In this section we have identified nine important repetitive risk factors which appear six times or more in over half of the selected papers that have been compiled and extracted. These factors are policy risk, preferential tariff rate, financing risk, technological development risk, market risk, projected investment profit, construction risk, risk of natural disasters, and partnership risk.

### 2.2. Discussion on Factors Affecting Investment Risks in Offshore Wind Power Industry

This section discusses a total of 15 documents, and the research and analysis reveal that there are 22 impact factors for partner selection, as shown in Table 2.

**Table 1.** Comparison table of investment risk influencing factors of the offshore wind power industry.

| Sn | Factor | Zhao et al. [37] | Masinia et al. [38] | Salo et al. [39] | Jin et al. [40] | Prostean et al. [41] | Balks et al. [42] | Lin Yanshuo [43] | Gatzert et al. [44] | Rolik et al. [45] | Dimitrios et al. [46] | Xu et al. [47] | Tu et al. [48] | Frq |
|---|---|---|---|---|---|---|---|---|---|---|---|---|---|---|
| 1 | Policy risk | ✓ | ✓ | ✓ | ✓ |  | ✓ | ✓ | ✓ | ✓ | ✓ |  | ✓ | 10 |
| 2 | Fair bidding policy | ✓ |  |  |  |  |  |  | ✓ |  | ✓ |  |  | 3 |
| 3 | Economic incentives | ✓ | ✓ |  |  |  |  |  |  | ✓ | ✓ |  | ✓ | 5 |
| 4 | Preferential tariff rate | ✓ |  | ✓ |  |  |  |  | ✓ | ✓ | ✓ |  | ✓ | 6 |
| 5 | Financing risk | ✓ |  | ✓ |  |  | ✓ | ✓ | ✓ | ✓ | ✓ | ✓ | ✓ | 9 |
| 6 | Financial incentives | ✓ |  |  |  |  |  |  |  | ✓ |  | ✓ |  | 3 |
| 7 | Technology development risk | ✓ | ✓ | ✓ | ✓ | ✓ | ✓ | ✓ | ✓ | ✓ | ✓ | ✓ | ✓ | 12 |
| 8 | Wind power specification | ✓ | ✓ |  | ✓ |  |  |  |  |  |  |  |  | 3 |
| 9 | Wind power training | ✓ |  |  |  |  |  |  |  |  |  | ✓ |  | 2 |
| 10 | Social risk |  | ✓ |  |  |  |  | ✓ |  |  | ✓ |  |  | 3 |
| 11 | Market risk |  | ✓ |  | ✓ | ✓ | ✓ | ✓ | ✓ | ✓ | ✓ | ✓ |  | 9 |
| 12 | Investor experience |  | ✓ |  |  |  |  | ✓ | ✓ |  |  |  | ✓ | 4 |
| 13 | Financial risk |  | ✓ | ✓ |  | ✓ | ✓ |  |  | ✓ |  |  | ✓ | 5 |
| 14 | Estimated investment profit |  | ✓ |  | ✓ | ✓ |  |  | ✓ | ✓ |  |  | ✓ | 6 |
| 15 | Construction risk |  |  | ✓ | ✓ | ✓ |  | ✓ | ✓ |  |  |  | ✓ | 6 |
| 16 | Country risk |  |  | ✓ |  |  |  | ✓ | ✓ | ✓ | ✓ |  |  | 5 |
| 17 | Risk of natural disasters |  |  | ✓ | ✓ | ✓ | ✓ | ✓ | ✓ |  |  |  |  | 6 |
| 18 | Competitive risk |  |  |  | ✓ | ✓ | ✓ |  |  |  | ✓ |  | ✓ | 5 |
| 19 | Logistics risk |  |  |  |  | ✓ |  | ✓ |  |  |  | ✓ | ✓ | 4 |
| 20 | Partnership risk |  |  |  |  | ✓ | ✓ | ✓ | ✓ | ✓ |  |  | ✓ | 6 |
| 21 | Contract risk |  |  |  |  | ✓ |  | ✓ | ✓ |  |  |  | ✓ | 4 |
| 22 | Construction period risk |  |  |  |  | ✓ | ✓ |  |  |  |  |  |  | 2 |
| 23 | Supply chain risk |  |  |  |  | ✓ |  |  |  | ✓ |  | ✓ | ✓ | 4 |
| 24 | Environmental risk |  |  |  |  |  |  | ✓ |  |  |  |  |  | 1 |
| 25 | Risk transfer-insurance |  |  |  |  |  |  |  | ✓ |  |  |  |  | 1 |
| 26 | Developer's credit risk |  |  |  |  |  |  |  | ✓ | ✓ |  |  | ✓ | 3 |

**Table 2.** Comparison of impact factors for partner selection.

| Sn | Factor | Bayazit et al. [49] | Eriksson et al. [50] | TPC 1 [51] | Mat et al. [52] | ROC CPAMI [53] | Watt et al. [54] | Horta et al. [55] | Liang et al. [56] | TPC 2 [57] | Cao Tingting [58] | Lin Hsinua [59] | Lu Weishien [60] | Liu et al. [61] | Neptune [62] | TPC3 [63] | Frq |
|---|---|---|---|---|---|---|---|---|---|---|---|---|---|---|---|---|---|
| 1 | Company reputation | ✓ | | | ✓ | | ✓ | ✓ | | | ✓ | | | ✓ | | ✓ | 7 |
| 2 | Track records | ✓ | ✓ | | ✓ | ✓ | ✓ | ✓ | ✓ | ✓ | ✓ | ✓ | ✓ | ✓ | ✓ | ✓ | 14 |
| 3 | Integration capabilities | ✓ | | | | | ✓ | | | | ✓ | | ✓ | | | | 5 |
| 4 | Technical ability | ✓ | ✓ | ✓ | ✓ | ✓ | ✓ | ✓ | ✓ | ✓ | ✓ | ✓ | ✓ | | ✓ | ✓ | 14 |
| 5 | Quality of staff | ✓ | ✓ | ✓ | | | ✓ | ✓ | | ✓ | ✓ | ✓ | | ✓ | | | 9 |
| 6 | Group culture | ✓ | | | ✓ | | | | | | | | | ✓ | | | 3 |
| 7 | Willingness to cooperate | ✓ | ✓ | ✓ | ✓ | | | | | | | | | ✓ | | ✓ | 6 |
| 8 | Risk management | ✓ | ✓ | ✓ | | | ✓ | | | ✓ | ✓ | | | | ✓ | | 7 |
| 9 | Performance ability | ✓ | | | ✓ | | | ✓ | ✓ | ✓ | ✓ | | ✓ | | ✓ | ✓ | 9 |
| 10 | Financial capability | ✓ | ✓ | ✓ | | ✓ | ✓ | ✓ | | | ✓ | ✓ | | ✓ | ✓ | ✓ | 11 |
| 11 | Market viability | ✓ | | ✓ | ✓ | | ✓ | | | | ✓ | ✓ | | ✓ | | | 8 |
| 12 | Management ability | ✓ | ✓ | | ✓ | ✓ | ✓ | ✓ | | ✓ | ✓ | | | ✓ | | | 9 |
| 13 | Pricing and cost | ✓ | | | | | ✓ | ✓ | | ✓ | ✓ | ✓ | | ✓ | ✓ | ✓ | 9 |
| 14 | Information sharing | | ✓ | | ✓ | | | | ✓ | | | ✓ | ✓ | ✓ | | ✓ | 7 |
| 15 | Research and innovation | | ✓ | | | ✓ | | ✓ | | ✓ | ✓ | ✓ | | ✓ | | ✓ | 8 |
| 16 | Organization size | | | ✓ | | ✓ | ✓ | | | ✓ | ✓ | | | | | | 5 |
| 17 | Ship quantity rerformance | | | ✓ | | | | | | ✓ | ✓ | ✓ | ✓ | | ✓ | ✓ | 7 |
| 18 | Negotiating ability | | | | ✓ | ✓ | | | | | | | | | | | 2 |
| 19 | Construction quality | | | | | ✓ | | ✓ | | ✓ | ✓ | | | ✓ | | ✓ | 6 |
| 20 | Safety management | | | | | | | ✓ | | ✓ | ✓ | | ✓ | | | ✓ | 6 |
| 21 | Port facility energy | | | | | | | | | | | | ✓ | | | | 1 |
| 22 | Communication ability | | | | | | | | | | | | | | | ✓ | 1 |

In this section we have identified 13 important repetitive risk factors which appear seven times or more in over half of the selected papers compiled and extracted. These factors are company reputation, track record/past performance, technical ability, quality of staff, risk management and resilience, ability to fulfill on schedule, financial capability, market viability, management ability, pricing and cost, information sharing, research and innovation, availability, and performance of ships and equipment.

### 2.3. Fuzzy Preference Relations (FPR)

The FPR proposed by Herrera-Viedma et al. (2004) improves the inconsistency issue caused by multi-decision-makers, multi-criteria, and multi-candidates in the AHP method. FPR not only simplifies the number of pairwise comparisons, but also brings ease to the calculation, which is a valuable reference in real-life applications [21].

Past cases show research on FPR implementation, project contracting strategy [64], the decision-making process of partner selection in international joint contract projects [65], decision making of subcontractor selection in the construction industry [66], decision making of general contractor selection projects [67], etc.

The following is an introduction to the FPR definition proposed by Herrera-Viedma et al. [20]:

3. Multiplicative Preference Relation (MPR):

After each plan is measured by the assessment scale, the preference relationship matrix A is established. Among them $A \in X \times X$, $A=(a_{ij})$, $(a_{ij}) \in \left[\frac{1}{9}, 9\right]$, $a_{ij}$ are the paired comparison preference values of the $X_i$ attribute and $X_i$ attribute, whereas $a_{ij}$ is the paired comparison preference value of the *i*-th attribute and the *j*-th attribute upon evaluation on the assessment scale. $a_{ij}$ and $a_{ji}$ have a reciprocal MPR relationship, namely: $a_{ij} \times a_{ji} = 1$.

4. Additive Preference Relationship:

Matrix *b* is composed of *X* attribute plan, where $b = (b_{ij})$, $b_{ij} = \mu_p(x_i, x_j)$, $b_{ij}$ is the pairwise preference comparison value of attribute $x_i$ and attribute $x_j$. When $b_{ij}$ and $b_{ji}$ add to equal 1, it is called the additive fuzzy preference relationship, as shown below: $b_{ij} + b_{ji} = 1 \; \forall i, j \in \{1, \ldots, n\}$.

Herrera-Viedma et al., proposed a method to convert the multiplicative preference relation matrix *A* into the fuzzy preference relation matrix *B*. This study refers to the advantages of applying the two. First, the semantic variables and the corresponding quantitative values are defined by the multiplicative preference relation and then the subjective opinions of each evaluator are collected through the questionnaire. After integrating the opinions of all experts to obtain the consistency weight formula calculation and conversion, the fuzzy preference relation is weighted by average.

### 2.4. Step-Wise Weight Assessment Ratio Analysis (SWARA)

In a MCDM problem, the calculation of the weight of the assessment criterion is of utmost importance for the selection or ranking process. There are three types of standard weighing methods developed in the past: objective, subjective, and aggregated methods. The subjective method is based on the expert's experience and personal judgment from implicit knowledge [31]. The SWARA method is a new subjective criteria assessment method, which is widely used in various fields such as economy, management, industry, manufacturing, design, architecture, policy, and environmental sustainability [32,33].

The SWARA method is a successful MCDM method developed by Keršuliene et al. (2010). Obtained from the knowledge and experience of the decision makers, it is widely used to determine the weight of evaluation criteria. It can assess the judgment or priority of more than one person in the binary comparison process. Assuming that every member of the group has the ability to judge on all criteria, these decisions should be combined to make compromises, in which case the team members can reach a consensus on the topic or combine different judgments or priorities with the geometric mean method. The SWARA

method used to determine the relative weight of the criteria can be accurately shown by the following steps [31,34]:

Step 1.  Determine the order of each criterion: Sort the influence and scores in descending order according to the importance of the conditions;

Step 2.  The relative importance of the criteria:

Step 3.  Recalculate the coefficient of weight; Starting from the second criterion, experts have pointed out that the ratio of standard *j* to the previous (*j-1*) attribute is called the relative importance of the average value and is expressed as *Sj*;

Step 4.  Calculate the initial weight shown as follows;

Step 5.  Calculate the final weight shown as follows.

*2.5. Fuzzy Technique for Order Preference by Similarity to Ideal Solution (FTOPSIS)*

TOPSIS is a MCDM method developed by Hwang and Yoon (1981). This method mainly tackles the definition and application of the positive ideal solution and the negative ideal solution The so-called positive ideal solution is the one with the greatest benefit criteria value and the smallest cost criteria value in each candidate; conversely, the negative ideal solution is the one with the smallest benefit criteria value and the greatest cost criteria value in each candidate. The basic principle is that, when choosing a plan, the best plan is the one closest to the positive ideal solution and furthest from the negative ideal solution [68].

In real life, the decision-maker's judgment may be ambiguous, as it depends on the uncertainty of human judgment and cannot be expressed with precise numerical values. Under these circumstances, the use of fuzzy set theory to model human judgments is advisable; this is called fuzzy multi criteria decision making (FMCDM) [69]. In order to avoid omissions or oversight caused by imprecise assessment data and human judgment, Chen (2000) proposed an extension of the TOPSIS model to a fuzzy environment, aiming to solve the problems of linguistic ambiguity and the need for collective decision-making as well as to eliminate the ambiguity caused by human judgment in the decision-making process [35].

The most significant feature of the FTOPSIS method is that it allows decision criteria to have different significant weights. Considering that each decision maker may have different weight preferences for various criteria, this method enables more accurate analysis and the most effective decision making [36].

The fuzzy preference sequence evaluation method proposed in this research uses k experts to use semantic variables to evaluate all candidate solutions under each criterion. There are m solutions ($Ai, i = 1, \ldots, m$) and n criteria ($Cj, j = 1, \ldots, n$). After the evaluation is completed, the geometric average method is used to integrate the opinions of all experts, then the gap between each candidate plan and the positive and negative ideal solution is calculated by the measurement method and, finally, the proximity coefficient of the candidate plan is calculated and the order of the ideal plan is obtained.

## 3. Establishing a Model for Investment Evaluation and Partner Selection in the Offshore Wind Power Foundations Industry

The construction model of this research can be divided into two decision-making stages (see Figure 1): (1) investment evaluation: assisting domestic manufacturers in the early evaluation of the risks of investing in offshore wind power underwater foundations industries; (2) partner selection: selecting the most suitable partnership to cooperate and participate in bidding to achieve the goal of localization of the offshore wind power industry and enhance the source of energy in the localized industry.

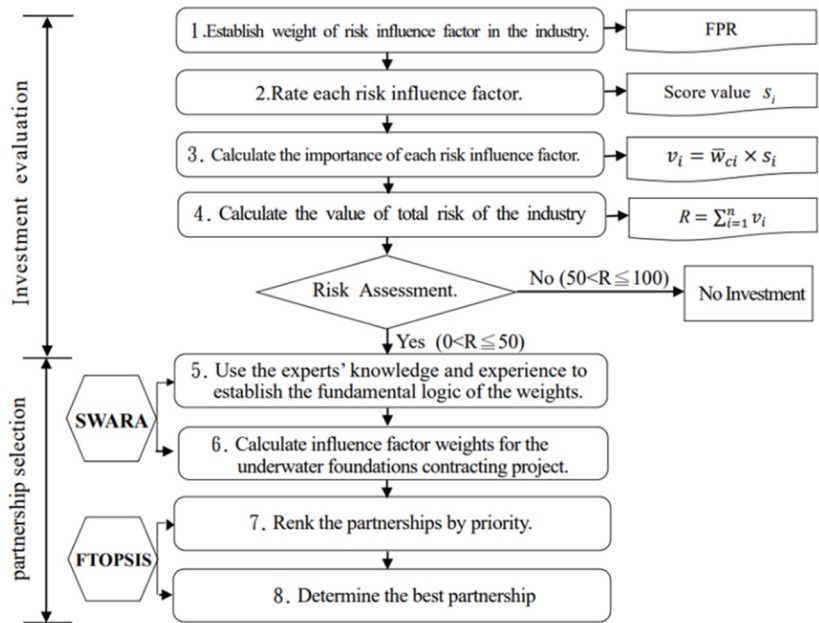

**Figure 1.** Illustrates the flow of the investment evaluation and partner selection model in the offshore wind power underwater foundations industry.

### 3.1. Investment Evaluation Model (IEM)

3.1.1. Evaluation Model Construction Process

Through literature review and compilation, the decision-making model of this research refers to domestic and foreign literatures in order to establish the preliminary influence factors; nine important repetitive risk factors from over half of the selected papers are cited as the decision-making influence factors. Upon obtaining the weight of each factor through questionnaire surveys and FPR calculation, the decision maker then adopts five different scales to rate the degree of risk preference based on past experience and existing data, and finally calculates the overall industry risk value in order to decide on whether to make the investment.

3.1.2. Calculating the Weights of Investment Risk Factors in the Offshore Wind Power Industry

The investment risk factors of the offshore wind power industry are identified in this section based on Table 1. The weight of each factor is calculated according to the questionnaire survey of experts and scholars. The steps and instructions are as follows:

Step 1.    Define linguistic variables:

This study cites and refers to the research results of E. Herrera-Viedma et al. (2004) [21] and defines the linguistic variable as (1/9, 9) as shown in Table 3.

Step 2.    Expert questionnaire:

**Table 3.** Linguistic variables to calculate the weights of influence factors.

| Linguistic Variables | Importance |
|---|---|
| Absolutely more important | 9 |
| Very strongly more important | 7 |
| Strongly more important | 5 |
| Weakly more important | 3 |
| Equally important | 1 |
| Weakly less important | 1/3 |
| Strongly less important | 1/5 |
| Very strongly less important | 1/7 |
| Absolutely less important | 1/9 |

During this stage, the objective is to ascertain the risks that underwater foundations manufacturers may face in the emerging market before they partake in the supply chain of the offshore wind power industry. This questionnaire only requires respondents to compare the relative importance of the listed influencing factors.

Step 3.   Build the MPR matrix:

Take the evaluation results of the *k*-th evaluator as an example to illustrate the conversion process. First, correlate the evaluation results to Table 3 and correlate the linguistic variable symbols of the evaluation to the quantitative value representing the evaluator's preference, and fill in the upper-right elements $\left(a_{12}^k, a_{23}^k, \dots\right)$ of the main diagonal line of the MPR matrix $A^k$ in the order as follows [21]:

$$
A^k = \begin{bmatrix}
1 & a_{12}^k & & & & & \\
 & 1 & a_{23}^k & & & & \\
 & & \ddots & \ddots & & & \\
 & & & 1 & a_{ij}^k & & \\
 & & & & \ddots & \ddots & \\
 & & & & & 1 & a_{(m-1)m}^k \\
 & & & & & & 1
\end{bmatrix} \tag{1}
$$

Step 4.   Establish the values of the other elements in the upper right corner of the MPR matrix:

Use Equation (2) to calculate the value of each element in the upper right corner diagonal to the preference relation (MPR) matrix $A^k$ [21]:

$$
a_{ij}^k = a_{i(j-1)}^k \times a_{(i+1)j}^k / a_{(i+1)(j-1)}^k \tag{2}
$$

Step 5.   Establish the value of the lower left element of the diagonal line of the MPR matrix $A^k$ [21]:

Calculate the value of the lower left part diagonal to the MPR matrix $A^k$ according to the following formula.

$$
a_{ij}^k = 1/a_{ji}^k \tag{3}
$$

The obtained MPR matrix $A^k$ is as follows:

$$
= \begin{bmatrix}
1 & a_{12}^k & a_{13}^k & \cdots & \cdots & a_{1m}^k \\
a_{21}^k & 1 & a_{23}^k & \cdots & \cdots & a_{2m}^k \\
a_{31}^k & a_{32}^k & 1 & \ddots & \cdots & \vdots \\
\vdots & \vdots & \ddots & \ddots & \ddots & \vdots \\
\vdots & \vdots & \vdots & \ddots & 1 & a_{(m-1)m}^k \\
a_{m1}^k & a_{m2}^k & \cdots & \cdots & a_{m(m-1)}^k & 1
\end{bmatrix} \tag{4}
$$

Step 6.　Find the maximum value $z^k$ in the MPR matrix $A^k$.

Step 7.　Convert the MPR matrix $A^k$ into a consistent MPR matrix $C^k$

Use Equation (20) to convert the MPR matrix $A^k$ into a consistent MPR matrix $C^k$ [21].

$$
c_{ij}^k = \left( a_{ij}^k \right)^{1/\log_9 z^k} \tag{5}
$$

Step 8.　Convert the consistent MPR matrix $C^k$ into the FPR matrix $B^k$. Convert the MPR matrix $C^k$, of which the value range is within (1/9,9). Use Equation (6) to convert to an FPR matrix $B^k$ with a value range within (0, 1) [21].

$$
b_{ij}^k = \left( 1 + \log_9 c_{ij}^k \right)/2 \tag{6}
$$

Step 9.　Average the respondents' pairwise matrix FPR:

Use Equation (7) to calculate the average FPR matrix $\overline{B}$ of $n$ evaluators [21].

$$
\overline{B} = \left( B^1 + \ldots + B^n \right)/n \tag{7}
$$

Step 10.　Normalize the FPR matrix obtained from step 9 [21].

Use Equation (8) to calculate the normalized average FPR matrix $R$, represented as $R = [r_{ij}]$

$$
r_{ij} = \overline{b}_{ij} / \sum_i \overline{b}_{ij} \tag{8}
$$

Step 11.　Obtain the weight of each factor:

As shown in Equation (9), the weight $\overline{w}_{ci}$ of each factor in the evaluation can be found [21].

$$
\overline{w}_{ci} = \frac{\sum_{j-1}^n r_{ij}}{\sum_{i=1}^n \sum_{j-1}^n r_{ij}} \tag{9}
$$

### 3.1.3. Rating the Risk Influence Factors

This study adopts the model proposed by Makarand Hasta et al. [70] to classify the degree of risk into five levels: (1) no risk, (2) low risk, (3) medium risk, (4) high risk, (5) absolute risk. A risk score is given according to the decision maker's risk preference as defined in Table 4.

**Table 4.** Risk level score card.

| Degree of Risk | Definition | Score |
|---|---|---|
| Lowest risk | Lowest probability, lowest severity | 0 |
| Lower risk | Lower probability, lower severity | 25 |
| Moderate risk | Moderate probability, moderate severity | 50 |
| Higher risk | Higher probability, higher severity | 75 |
| Highest risk | Highest probability, highest severity | 100 |

Given the above evaluation criteria, risk influence factors are scored based on the decision maker's past experience and existing data.

Based on to Equation (10), calculate the weighted average score $\overline{w}_{ci}$, $i = 1, 2, \ldots, n$, by calculating the weighted value of each influencing factor in Section 3.1.2 and the average score of the different evaluation factors as filled in by the experts in this section [65].

$$v_i = \overline{w}_{ci} \times s_i \tag{10}$$

Individually obtain the product of the influence factor weight value and the average score value; further, obtain the weighted average score of each factor $v_i$, $i = 1, 2, \ldots, n$.

Calculate the weighted average score of each factor ($v_i$, $i = 1, 2, \ldots, n$.) from Equation (10) and add the weighted average score of each factor $v_i$ to calculate the overall industry investment risk value $R$, as shown in Equation (11) [65].

$$R = \sum_{i=1}^{n} v_i, i = 1, 2, \ldots, n \tag{11}$$

At present, the development of Taiwan's offshore wind power industry is still in the early stage. For the local manufacturers with offshore wind power underwater foundations capabilities, whether to take part in the investment of the offshore wind power supply chain is an important, but difficult judgement to make. This research seeks to be prudent and thus formulates the risk thresholds for investment in the industry as follows:

Risk value between $0 < R \leq 50$: risk is low and investment in the offshore wind power industry is viable.

Risk value between $50 < R \leq 100$: investment risk is high. Corporations are advised not to enter the offshore wind power market at this moment.

### 3.2. Partner Selection Model (PSM)

Upon completion of the preliminary investment risk assessment in Section 3.1.3, the risk value $0 < R \leq 50$ is obtained, which qualifies for subsequent project partner selection. Through literature review, we compile and refer to domestic and foreign literatures and establish the preliminary influence factors. In over half of the papers, we compiled and extracted the decision influence factors; we have identified and cited repetitive risk factors. We use questionnaire survey and the SWARA method to calculate the weight of each factor and the FTOPSIS method to select the most suitable partner.

#### 3.2.1. Calculate the Weight of the Influence Factors

At this stage, the SWARA method is adopted to establish the weight of the influence factors. The steps are as follows:

Step 1.   Importance ranking:

The experts fill in the evaluation results in Table 5 and rank the order of importance from 1 (most important) to 13 (least important).

Step 2.   Importance comparison:

**Table 5.** SWARA weight calculation of the influence factors.

| Item | Influence Factor | Importance Ranking | $s_j$ | $k_j$ | $q_j$ | $w_j$ |
|------|------------------|:------------------:|:-----:|:-----:|:-----:|:-----:|
| F1 | Company Reputation | - | - | - | - | - |
| F2 | Track Record/Past Performance | - | - | - | - | - |
| F3 | Technical Ability | - | - | - | - | - |
| F4 | Quality of Staff | - | - | - | - | - |
| F5 | Risk Management and Resilience | - | - | - | - | - |
| F6 | Ability to Fulfill on Schedule | - | - | - | - | - |
| F7 | Financial Capability | - | - | - | - | - |
| F8 | Market Viability | - | - | - | - | - |
| F9 | Management Ability | - | - | - | - | - |
| F10 | Pricing and Cost | - | - | - | - | - |
| F11 | Information Sharing | - | - | - | - | - |
| F12 | Research and Innovation | - | - | - | - | - |
| F13 | Availability and Performance of Ships and Equipment | - | - | - | - | - |

Ask the expert to compare the importance of factor *j* as ranked in Step 1 with the previous (*j*-1) attribute. Ranking 1 is compared with Ranking 2, and Ranking 2 is compared with Ranking 3, and so forth. In this study, the importance ratio of each factor is set as a multiple of 5%, indicating its importance relative to other factors (when filling in, 5% is represented by 0.05, 10% is 0.10, and 25% is 0.25). This ratio is called the relative importance of the average, and is expressed as $s_j$ .

Step 3.   Calculate the relative importance function $k_j$ according to Formula (12) [34]:

$$k_j = \begin{cases} 1 & j = 1 \\ s_j + 1 & j > 1 \end{cases} \tag{12}$$

Step 4.   Calculate the initial weight $q_j$ according to Formula (13) [34]:

$$q_j = \begin{cases} 1 & j = 1 \\ \frac{q_{j-1}}{k_j} & j > 1 \end{cases} \tag{13}$$

Step 5.   Obtain the final weight $w_j$ according to Formula (14) [34]:

$$w_j = \frac{q_j}{\sum_{k=1}^{n} q_j} \tag{14}$$

where $w_j$ represents the relative weight of condition *j*.

### 3.2.2. Establish the Most Suitable Partnership Ranking

Following the influence factor weights obtained in Section 3.2.1, use FTOPSIS as introduced in Section 2.5 in the literature review. After calculating the ranking of candidates in this section, determine the most suitable partnership in the order of candidates obtained. The steps and instructions are as follows:

Step 1.   Use of linguistic variables to evaluate the candidates:

Each member of the decision-making team is asked to evaluate the listed 13 influencing factors and potential partners with the scale in Table 6. Let $m$ partnerships be represented by the set $P = \{P_1, P_i, \ldots, P_m\}$. Assuming that each candidate has $k$ evaluation criteria, it is represented by the set $F = \{F_1, F_i, \ldots, F_k\}$. Suppose there are $l$ decision-makers in a decision-making team (represented by $D_1, D_2, \ldots, D_m$), and the $r$-th decision maker gives a fuzzy linguistic scale to the score that the $i$-th candidate obtains under the $j$-th criterion, then the corresponding triangular fuzzy number is set $\tilde{x}_{ijr} = a_{ijr}, b_{ijr}, c_{ijr}, i = 1, 2, \ldots, m; j = 1, 2, \ldots, k; r = 1, 2, \ldots, l$.

Step 2.  Establish fuzzy decision matrix:

**Table 6.** Grade of linguistic variables in FTOPSIS.

| Linguistic the Variables | Triangular Fuzzy Numbers |
|---|---|
| Very poor (VP) | (0, 0, 1) |
| Poor (P) | (0, 1, 3) |
| Mildly poor (MP) | (1, 3, 5) |
| Fair (F) | (3, 5, 7) |
| Mildly good (MG) | (5, 7, 9) |
| Good (G) | (7, 9, 10) |
| Very good (VG) | (9, 10, 10) |

Use the aggregated triangular fuzzy number obtained by the $r$-th decision maker for the $i$-th alternative under the $j$-th criterion [69].

$$\tilde{x}_{ijr} = a_{ijr}, b_{ijr}, c_{ijr}, i = 1, 2, \ldots, m; j = 1, 2, \ldots, k; r = 1, 2, \ldots, l \qquad (15)$$

to obtain the aggregated triangular fuzzy number

$$\tilde{x}_{ij} = a_{ij}, b_{ij}, c_{ij}, i = 1, 2, \ldots, m; j = 1, 2, \ldots, k$$

where $a_{ij} = \min\limits_{1 \leq r \leq 1} \{a_{ijr}\}, b_{ij} = \frac{1}{i} \sum_{r=1}^{i} b_{ijr}, c_{ij} = \max\limits_{1 \leq r \leq 1} \{c_{ijr}\}$.

Use $\tilde{x}_{ij}$ as obtained to construct the fuzzy decision matrix $\tilde{D}$ [69]:

$$F_1 \quad \cdots \quad F_j \quad \cdots \quad F_k \tilde{D} \equiv [\tilde{x}_{ij}]_{m \times k} = \begin{matrix} P_1 \\ \vdots \\ P_i \\ \vdots \\ P_m \end{matrix} \begin{bmatrix} \tilde{x}_{11} & \cdots \tilde{x}_{1j} \cdots & \tilde{x}_{1k} \\ \vdots & \vdots & \vdots \\ \tilde{x}_{i1} & \cdots \tilde{x}_{ij} \cdots & \tilde{x}_{ik} \\ \vdots & \vdots & \vdots \\ \tilde{x}_{m1} & \cdots \tilde{x}_{mj} \cdots & \tilde{x}_{mk} \end{bmatrix} \qquad (16)$$

Step 3.  Establish a normalized fuzzy matrix:

Let $B$ denote the benefit criteria and $C$ denote the cost criteria.

Pair $j = 1, 2, \ldots, k$

If $F_j \in B$, then $\tilde{r}_{ij} = \frac{1}{c_j^+}(\times)\tilde{x}_{ij} = \left(\frac{a_{ij}}{c_j^+}, \frac{b_{ij}}{c_j^+}, \frac{c_{ij}}{c_j^+}\right)$, among them $c_j^+ = \max\limits_{1 \leq i \leq 1} c_{ij}$. If $F_j \in C$,

then $\tilde{r}_{ij} = a_j^-(\times)\tilde{x}_{ij}^{-1} = \left(\frac{a_j^-}{c_{ij}}, \frac{a_j^-}{b_{ij}}, \frac{a_j^-}{a_{ij}}\right)$, among them $a_j^- = \min\limits_{1 \leq i \leq 1} a_{ij}$.

Thus, a normalized fuzzy decision matrix $\widetilde{G}$ is obtained as follows [69]:

$$
F_1 \cdots F_j \cdots F_k \widetilde{G} \equiv \left[\widetilde{r}_{ij}\right]_{m\times k} = 
\begin{array}{c}
P_1 \\ \vdots \\ P_i \\ \vdots \\ P_{\phantom{m}m}
\end{array}
\left[
\begin{array}{cccc}
\widetilde{r}_{11} & \cdots & \widetilde{r}_{1j} \cdots & \widetilde{r}_{1k} \\
\vdots & & \vdots & \vdots \\
\widetilde{r}_{i1} & \cdots \widetilde{r}_{ij} \cdots & & \widetilde{r}_{ik} \\
\vdots & & \vdots & \vdots \\
\widetilde{r}_{m1} & \cdots & \widetilde{r}_{mj} \cdots & \widetilde{r}_{mk}
\end{array}
\right]
\tag{17}
$$

Step 4.  Calculate the weighted normalized fuzzy decision matrix:

Add vector weight to each row of the standardized fuzzy decision matrix $\widetilde{G}$ (SWARA); the fuzzy weights are $\widetilde{w}_1, \widetilde{w}_2, \ldots \widetilde{w}_k$ respectively, and a new matrix $\widetilde{T}$ is obtained as follows [69]:

$$
\widetilde{T} \equiv \left[\widetilde{v}_{ij}\right]_{m\times k} = 
\left[
\begin{array}{cccc}
\widetilde{v}_{11} & \cdots & \widetilde{v}_{1j} \cdots & \widetilde{v}_{1k} \\
\vdots & & \vdots & \vdots \\
\widetilde{v}_{i1} & \cdots \widetilde{v}_{ij} \cdots & & \widetilde{v}_{ik} \\
\vdots & & \vdots & \vdots \\
\widetilde{v}_{m1} & \cdots & \widetilde{v}_{mj} \cdots & \widetilde{v}_{mk}
\end{array}
\right]
\tag{18}
$$

where $v_{ij} = w_j(\times)r_{ij} = \left(\varphi_{ij}, \zeta_{ij}, \omega_{ij}\right), i = 1, 2, \ldots, m; j = 1, 2, \ldots, k$

Step 5.  Calculate the fuzzy positive ideal solution $V^+$ and fuzzy negative ideal solution $V^-$ with Equations (19) and (20):

The positive ideal solution is set as [69]

$$
V^+ = \left(\widetilde{v}_1^+, \widetilde{v}_2^+, \ldots \widetilde{v}_k^+\right) \text{where } \widetilde{v}_j^+ = \max_i\{v_{ij3}\}
\tag{19}
$$

The negative ideal solution is set as [69]

$$
V^- = \left(\widetilde{v}_1^-, \widetilde{v}_2^-, \ldots \widetilde{v}_k^-\right) \text{where } \widetilde{v}_j^- = \min_i\{v_{ij1}\}
\tag{20}
$$

Step 6.  Use the measurement method to find the gap between each candidate and the positive and negative ideal solutions [69]:

$$
d_i^+ = \sum_{j=1}^n d\left(\widetilde{v}_{ij}, \widetilde{v}_j^+\right), d_i^- = \sum_{j=1}^n d\left(\widetilde{v}_{ij}, \widetilde{v}_j^-\right), i = 1, 2 \ldots, m,
\tag{21}
$$

where $d(\widetilde{m}, \widetilde{n})$ represents the measured distance of two fuzzy numbers. Assuming two triangular fuzzy numbers $\widetilde{m} = m_1, m_2, m_3$ and $\widetilde{n} = n_1, n_2, n_3$, the measured distance between the two is calculated as follows [69]:

$$
d\widetilde{m}, \widetilde{n} = \sqrt{\frac{1}{3}\left[m_1 - n_1^2 + m_2 - n_2^2 + m_3 - n_3^2\right]}
\tag{22}
$$

Step 7.  Calculate the correlation coefficient (*CCi*) and rank the order of the alternatives:

By calculating the *CCi* value of each plan by Formula (23), the ranking order of all partnerships is thus determined. The one with the highest *CCi* value is the best choice [69].

$$
CC_i = \frac{d_i^-}{d_i^- + d_i^+}, i = 1, 2 \ldots, m,
\tag{23}
$$

## 4. Case Analysis and Verification of Decision-Making Model

In this chapter, the decision model in the case of Taipower's Offshore Wind Power Generation Phase II, a wind farm construction project, is verified. First, the investment risk from the standpoint of an underwater infrastructure builder is assessed and then the best partnership is selected among the three constructor candidates who possess the capabilities to undertake underwater infrastructure transportation and installation in Taiwan is and can jointly participate in the bidding. The contents are as follows: (1) case background, (2) preliminary stage of investment evaluation, (3) project partner selection stage, (4) establishing the best decision-making plan.

### 4.1. Case Background

4.1.1. The Case of Offshore Wind Farm Development

The study examines the preliminary investment evaluation of offshore wind power underwater foundations. It verifies the decision model in the case of Taipower's Offshore Wind Power Generation Phase II, a wind farm construction project, as an effort to provide useful reference for domestic manufacturers of underwater foundations to conduct evaluation in the early stage of their investment decision. The introduction of the case is as follows:

1.  Project content: The total installation capacity of this project is 300 MW. Single-unit capacities of 5.2 MW, 6 MW, 8 MW, and 9.5 MW wind turbines are planned. Individual wind turbines are planned with structural support from the bottom of the sleeves (jacket-type).
2.  Project location: Located in the western offshore waters of Lugang District, Changbin Industrial Zone, Changhua County, the closest distance from the shore is about 16 km; the water depth is about 37–49 m.
3.  Project benefits: The financial assessment of the cost benefit analysis of this project is passed. The results reflect the growing trend of large-scale wind turbines and the anticipated positive results of the cost benefit analysis.
4.  Project contracting (wind turbine substructure): The contracting is expected to start in April 2021 for a 3-month duration and to be completed by the end of June 2021.
5.  Preparatory work for the lower structure of the wind turbines: The preparatory work is expected to start in July 2021 for a 9-month duration and to be completed by the end of March 2022.
6.  Production of the wind turbine substructure: The production is expected to start in April 2022. Preliminary assessment indicates that a sleeve (jacket-type) substructure takes about 10 to 14 days to manufacture, so 50 sleeve substructures can be produced per year on average. Based on this production rate, completion can be expected by the end of March 2024.
7.  Construction of the wine turbine substructure: Installation is planned to be carried out from April to September every year, to avoid the northeast monsoon season. The construction is scheduled to start in April 2023 and the installation of all the wind turbine substructures will be completed by the middle of 2024.
8.  Project cost: The amount of investment of each plan is shown in Table 7.

4.1.2. Profiles of Potential Partners in the Underwater Foundations Industry

The profiles of the three manufacturers capable of contracting transporting and constructing offshore wind power underwater foundations in Taiwan are summarized herein, as shown in Table 8.

**Table 7.** Investment budget in the case (amount in USD ).

|  | Underwater Foundations | Piles | Underwater Structure Installation |
|---|---|---|---|
| 5.2 MW (57 units) | 230 million | 150 million | 130 million |
| 6 MW (50 units) | 300 million | 140 million | 140 million |
| 8 MW (37 units) | 240 million | 90 million | 90 million |
| 9.5 MW (31 units) | 260 million | 80 million | 90 million |

**Table 8.** Partner Profiles.

| Number | Company Features |
|---|---|
| P1 | 1. P1 is a world renowned offshore wind farm marine engineering and construction company with turnkey experience and technology. It was born of a merger between GeoSea (DEME Group) and Taiwan's largest shipbuilding company, providing construction and maintenance services for offshore wind power industrial vessels. To complay with the localization policy, the company can mobilize domestic fleets, including large barges and tugboats. It can also meet the transporting and installation capacity of offshore wind farms in Taiwan. 2. P1 is the first company in Taiwan that provides engineering, procurement, construction, and installation services. It imports GeoSea's marine engineering technology and management experience, and dispatches GeoSea's fleet for support. 3. GeoSea has successfully completed 73 offshore wind power projects. |
| P2 | 1. Established more than 40 years ago, P2 Construction Company has a proven track record of marine engineering and ships. In compliance with the government's new energy projects and industrial localization policy, a subsidiary was established, which joins in cooperation with the UK's High Speed Transfers (HST) for training and SMS management system technology transfer. 2. In 2019, five more offshore wind turbines were purchased and the fleet became 100% Taiwanese, which met the government's expectations of localization. An investment of 30 million US dollars is planned in the next 5 years for the introduction and setup of a cabin simulation training center, a platform to train more local crews via simulation and practice. 3. With more than 300 employees and 100 full-time crew members, over 40 work vessels, and the largest domestic marine engineering fleet, P2 Construction Company is the largest maritime construction engineering company in Taiwan. |
| P3 | 1. P3 Maritime Engineering is a joint venture company established by Boskalis, the world's largest maritime engineering company, and a local Taiwanese construction company. 2. With over 100 years of maritime engineering experience, more than 700 professional ships, and over 9500 professionals with a track record on over 100 offshore wind farm projects worldwide, Boskalis has 30 years of on-site maritime cooperation with P3 Construction in Taiwan and the Asia-Pacific region. 3. Through Boskalis' experience with many successful offshore wind power projects in Europe, along with P3's local maritime engineering knowledge in Taiwan, the partnership is able to offer services to Taiwan's wind farm projects such as engineering design; engineering, procurement, construction and installation (EPCI); manufacturing; transportation and installation; seabed renovation; maritime survey, wind farms decommissioning; etc. |

*4.2. Preliminary Investment Assessment*

According to the steps detailed in Section 3.1, companies must carefully review the various influencing factors, and evaluate the industrial risk impact factors according to the decision maker's personal risk preference to decide whether to make the investment.

### 4.2.1. FPR Weight Calculation of Risk Impact Factors

By means of questionnaires, this study surveys the weight of the industrial risk impact factors with respect to contractor investment. The respondents include experts from local consulting firms as well as engineers in the offshore wind power field. On average, the respondents have 7.2 years of work experience, among whom three hold doctorate degrees, fourteen hold master degrees, and one holds a bachelor degree.

Step 1.    Questionnaire survey:

Based on the definition of linguistic variable symbols in Table 4, 18 experts were asked to fill in the relative importance of the pairwise comparison of the influence factors. For example, the evaluation result of Expert 1 is shown in Table 9. The evaluation of the linguistic variable symbols corresponds with the quantified value that represents the evaluator's preference; the multiplicative preference relation (MPR) matrix in the upper right corner of the main diagonal is filled in, as shown in Table 10.

Step 2.    Establish MPR matrix A according to the quantitative value of the evaluation result:

**Table 9.** Linguistic variable evaluation of Expert 1.

| Evaluation Factor | Absolutely More Important | Very Strongly More Important | Strongly More Important | Weakly More Important | Equally Important | Weakly Less Important | Strongly Less Important | Very Strongly Less Important | Absolutely Less Important | Evaluation Factor |
|---|---|---|---|---|---|---|---|---|---|---|
| Policy risk | | | | | | | ✓ | | | Preferential tariff rate |
| Preferential tariff rate | | | | ✓ | | | | | | Financing risk |
| Financing risk | | ✓ | | | | | | | | Technological R&D risk |
| Technological R&D risk | | | | | ✓ | | | | | Market risk |
| Market risk | | | | | | | | | ✓ | projected investment profit |
| projected investment profit | | | | | ✓ | | | | | Construction risk |
| Construction risk | | | | ✓ | | | | | | Natural Disaster |
| Natural Disaster | | | | | ✓ | | | | | Financing risk |

**Table 10.** Quantified value of Expert 1's preference.

|  | CF1 | CF2 | CF3 | CF4 | CF5 | CF6 | CF7 | CF8 | CF9 |
|---|---|---|---|---|---|---|---|---|---|
| CF1 | 1.00 | 0.20 | | | | | | | |
| CF2 | | 1.00 | 3.00 | | | | | | |
| CF3 | | | 1.00 | 5.00 | | | | | |
| CF4 | | | | 1.00 | 1.00 | | | | |
| CF5 | | | | | 1.00 | 0.14 | | | |
| CF6 | | | | | | 1.00 | 1.00 | | |
| CF7 | | | | | | | 1.00 | 3.00 | |
| CF8 | | | | | | | | 1.00 | 1.00 |
| CF9 | | | | | | | | | 1.00 |

Use Equations (2)–(4) to calculate the value of each element to obtain the MPR matrix, as shown in Table 11.

Step 3. Calculate the consistent MPR matrix:

**Table 11.** MPR Matrix.

|  | CF1 | CF2 | CF3 | CF4 | CF5 | CF6 | CF7 | CF8 | CF9 |
|---|---|---|---|---|---|---|---|---|---|
| CF1 | 1.00 | 0.20 | 0.60 | 3.00 | 3.00 | 0.43 | 0.43 | 1.29 | 1 |
| CF2 | 5.00 | 1.00 | 3.00 | 15.00 | 15.00 | 2.14 | 2.14 | 6.43 | 6 |
| CF3 | 1.67 | 0.33 | 1.00 | 5.00 | 5.00 | 0.71 | 0.71 | 2.14 | 2.14 |
| CF4 | 0.33 | 0.07 | 0.20 | 1.00 | 1.00 | 0.14 | 0.14 | 0.43 | 0.43 |
| CF5 | 0.33 | 0.07 | 0.20 | 1.00 | 1.00 | 0.14 | 0.14 | 0.43 | 0.43 |
| CF6 | 2.33 | 0.47 | 1.40 | 7.00 | 7.00 | 1.00 | 1.00 | 3.00 | 3.00 |
| CF7 | 2.33 | 0.47 | 1.40 | 7.00 | 7.00 | 1.00 | 1.00 | 3.00 | 3.00 |
| CF8 | 0.78 | 0.16 | 0.47 | 2.33 | 2.33 | 0.33 | 0.33 | 1.00 | 1.00 |
| CF9 | 0.78 | 0.16 | 0.47 | 2.33 | 2.33 | 0.33 | 0.33 | 1.00 | 1.00 |

Convert MPR matrix *A* to MPR matrix *C* according to Equation (5), which is consistent, as shown in Table 12.

Step 4. Convert to FPR matrix:

**Table 12.** Consistent MPR Matrix.

|  | CF1 | CF2 | CF3 | CF4 | CF5 | CF6 | CF7 | CF8 | CF9 |
|---|---|---|---|---|---|---|---|---|---|
| CF1 | 1.00 | 0.27 | 0.66 | 2.44 | 2.44 | 0.50 | 0.50 | 1.23 | 1.23 |
| CF2 | 3.69 | 1.00 | 2.44 | 9.00 | 9.00 | 1.86 | 1.86 | 4.53 | 4.53 |
| CF3 | 1.51 | 0.41 | 1.00 | 3.69 | 3.69 | 0.76 | 0.76 | 1.86 | 1.86 |
| CF4 | 0.41 | 0.11 | 0.27 | 1.00 | 1.00 | 0.21 | 0.21 | 0.50 | 0.50 |
| CF5 | 0.41 | 0.11 | 0.27 | 1.00 | 1.00 | 0.21 | 0.21 | 0.50 | 0.50 |
| CF6 | 1.99 | 0.54 | 1.31 | 4.85 | 4.85 | 1.00 | 1.00 | 2.44 | 2.44 |
| CF7 | 1.99 | 0.54 | 1.31 | 4.85 | 4.85 | 1.00 | 1.00 | 2.44 | 2.44 |
| CF8 | 0.82 | 0.22 | 0.54 | 1.99 | 1.99 | 0.41 | 0.41 | 1.00 | 1.00 |
| CF9 | 0.82 | 0.22 | 0.54 | 1.99 | 1.99 | 0.41 | 0.41 | 1.00 | 1.00 |

Use Equation (6) to convert the consistent MPR matrix C into FPR matrix B, as shown in Table 13.

Step 5. Average the respondents' paired FPR matrices:

**Table 13.** FPR Matrix.

|  | CF1 | CF2 | CF3 | CF4 | CF5 | CF6 | CF7 | CF8 | CF9 |
|---|---|---|---|---|---|---|---|---|---|
| CF1 | 0.50 | 0.20 | 0.41 | 0.70 | 0.70 | 0.34 | 0.34 | 0.55 | 0.55 |
| CF2 | 0.80 | 0.50 | 0.70 | 1.00 | 1.00 | 0.64 | 0.64 | 0.84 | 0.84 |
| CF3 | 0.59 | 0.30 | 0.50 | 0.80 | 0.80 | 0.44 | 0.44 | 0.64 | 0.64 |
| CF4 | 0.30 | 0.00 | 0.20 | 0.50 | 0.50 | 0.14 | 0.14 | 0.34 | 0.34 |
| CF5 | 0.30 | 0.00 | 0.20 | 0.50 | 0.50 | 0.14 | 0.14 | 0.34 | 0.34 |
| CF6 | 0.66 | 0.36 | 0.56 | 0.86 | 0.86 | 0.50 | 0.50 | 0.70 | 0.70 |
| CF7 | 0.66 | 0.36 | 0.56 | 0.86 | 0.86 | 0.50 | 0.50 | 0.70 | 0.70 |
| CF8 | 0.45 | 0.16 | 0.36 | 0.66 | 0.66 | 0.30 | 0.30 | 0.50 | 0.50 |
| CF9 | 0.45 | 0.16 | 0.36 | 0.66 | 0.66 | 0.30 | 0.30 | 0.50 | 0.50 |

Obtain 18 FPR matrices according to the results of the 18 experts' questionnaires. Use Equation (7) to obtain the FPR matrix. The results are shown in Table 14.

Step 6.　Obtain the weight of each risk influence factor:

**Table 14.** Average FPR matrix.

|  | CF1 | CF2 | CF3 | CF4 | CF5 | CF6 | CF7 | CF8 | CF9 |
|---|---|---|---|---|---|---|---|---|---|
| CF1 | 0.500 | 0.704 | 0.567 | 0.673 | 0.633 | 0.585 | 0.565 | 0.619 | 0.573 |
| CF2 | 0.296 | 0.500 | 0.363 | 0.468 | 0.429 | 0.381 | 0.361 | 0.415 | 0.369 |
| CF3 | 0.433 | 0.637 | 0.500 | 0.606 | 0.567 | 0.518 | 0.498 | 0.552 | 0.506 |
| CF4 | 0.327 | 0.532 | 0.394 | 0.500 | 0.461 | 0.413 | 0.392 | 0.447 | 0.400 |
| CF5 | 0.367 | 0.571 | 0.433 | 0.539 | 0.500 | 0.452 | 0.431 | 0.486 | 0.439 |
| CF6 | 0.415 | 0.619 | 0.482 | 0.587 | 0.548 | 0.500 | 0.480 | 0.534 | 0.488 |
| CF7 | 0.435 | 0.639 | 0.502 | 0.608 | 0.569 | 0.520 | 0.500 | 0.554 | 0.508 |
| CF8 | 0.381 | 0.585 | 0.448 | 0.553 | 0.514 | 0.466 | 0.446 | 0.500 | 0.454 |
| CF9 | 0.427 | 0.631 | 0.494 | 0.600 | 0.561 | 0.512 | 0.492 | 0.546 | 0.500 |
| TOTAL | 3.118 | 4.921 | 4.352 | 5.410 | 5.232 | 5.107 | 4.066 | 4.240 | 4.053 |

Normalize the average FPR matrix obtained in Step 5 according to Formula (8) and calculate the weight of each risk factor with Formula (9). The results are shown in Table 15.

**Table 15.** Normalized average FPR matrix.

|  | CF1 | CF2 | CF3 | CF4 | CF5 | CF6 | CF7 | CF8 | CF9 | Subtotal | Relative Weights |
|---|---|---|---|---|---|---|---|---|---|---|---|
| CF1 | 0.160 | 0.143 | 0.130 | 0.124 | 0.121 | 0.115 | 0.139 | 0.146 | 0.141 | 1.220 | 0.134 |
| CF2 | 0.095 | 0.102 | 0.083 | 0.087 | 0.082 | 0.075 | 0.089 | 0.098 | 0.091 | 0.801 | 0.088 |
| CF3 | 0.139 | 0.129 | 0.115 | 0.112 | 0.108 | 0.102 | 0.122 | 0.130 | 0.125 | 1.083 | 0.119 |
| CF4 | 0.105 | 0.108 | 0.091 | 0.092 | 0.088 | 0.081 | 0.096 | 0.105 | 0.099 | 0.866 | 0.095 |
| CF5 | 0.118 | 0.116 | 0.100 | 0.100 | 0.096 | 0.088 | 0.106 | 0.115 | 0.108 | 0.946 | 0.104 |
| CF6 | 0.133 | 0.126 | 0.111 | 0.109 | 0.105 | 0.098 | 0.118 | 0.126 | 0.120 | 1.045 | 0.115 |
| CF7 | 0.140 | 0.130 | 0.115 | 0.112 | 0.109 | 0.102 | 0.123 | 0.131 | 0.125 | 1.087 | 0.120 |
| CF8 | 0.122 | 0.119 | 0.103 | 0.102 | 0.098 | 0.091 | 0.110 | 0.118 | 0.112 | 0.975 | 0.107 |
| CF9 | 0.137 | 0.128 | 0.114 | 0.111 | 0.107 | 0.100 | 0.121 | 0.129 | 0.123 | 1.070 | 0.118 |
|  |  |  |  |  |  |  |  |  |  | Total | |
|  |  |  |  |  |  |  |  |  |  | 9.092 | 1.000 |

Through the steps above, the weights of objective investment risk influence factors are established.

4.2.2. Rating the Influence Factors of Industrial Investment Risk

At this stage, three experts (two doctorates and one master's degree, with an average age of 15 years of experience) were invited to score the impact factors according to the definition in Table 4. The scoring results of the three experts are shown in Table 16.

**Table 16.** Ratings of investment risk influence factors.

| Risk Impact Factor | Expert 1 Score | Expert 2 Score | Expert 3 Score | Average |
|---|---|---|---|---|
| Policy risk | 25 | 50 | 25 | 33.333 |
| Preferential tariff rate | 25 | 25 | 25 | 25 |
| Financing risk | 75 | 50 | 50 | 58.333 |
| Technological development risk | 50 | 50 | 25 | 41.667 |
| Market risk | 50 | 50 | 25 | 41.667 |
| Projected investment profit | 50 | 25 | 50 | 41.667 |
| Construction risk | 75 | 75 | 75 | 75 |
| Natural disaster risk | 50 | 50 | 75 | 58.333 |
| Partnership risk | 25 | 25 | 25 | 25 |

By cross calculating the weighted value of each influence factor interactively in Section 4.2.1 and the average scores of the different evaluation factors as filled in by the experts in this section, the weighted average score $vi$ is obtained according to Equation (10). Equation (11) is used to add up the scores of all the factors to calculate the overall investment risk value R, as shown in Table 17.

**Table 17.** Assessment of overall industry investment risk.

| Risk Influence Factor | Factor Weight (W$ci$) | Risk Score ($si$) | Importance ($vi$) |
|---|---|---|---|
| Policy risk | 0.134 | 33.333 | 4.467 |
| Preferential tariff rate | 0.088 | 25.000 | 2.200 |
| Financing risk | 0.119 | 58.333 | 6.942 |
| Technological development risk | 0.095 | 41.667 | 3.958 |
| Market risk | 0.104 | 41.667 | 4.333 |
| Projected investment profit | 0.115 | 41.667 | 4.792 |
| Construction risk | 0.120 | 75.000 | 9.000 |
| Natural risk | 0.107 | 58.333 | 6.242 |
| Partner risk | 0.118 | 25.000 | 2.950 |
| Overall investment risk value *R* | | | 44.884 |

The industry's overall risk value of investment obtained in this case is 44.884. According to the definition stated in Section 3.1.3, the risk value is $0 < R \leq 50$, which means that the investment risk is within the permitted range and investment in the offshore wind power market is feasible. This research therefore moves onto the next stage: the decision-making model of project partner selection.

*4.3. Project Partner Selection Stage*

At this stage, the SWARA method is adopted to calculate the weight of each factor and the FTOPSIS method is used to select the most suitable partner.

4.3.1. Calculate the Weight Value of the Selection Influence Factor

At this stage, four experts (three doctorates and 1 master's degree, with an average age of 12 years of experience) were invited to score the factors according to the steps defined in Section 3.2.1.

Step 1.   Importance ranking and importance comparison:

In Table 18, Expert 1 is shown as a rating example of the new ranking and importance comparison $Sj$; the same method also applies to the other 3 experts.

Step 2.   Calculate the relative importance function $k_j$ according to Formula (12), as shown in Table 19.

**Table 18.** Expert 1's new ranking and importance comparison of influence factors.

| Item | Influence Factor | Importance Ranking | New Ranking | S$j$ |
|---|---|---|---|---|
| F1 | Company Reputation | 13 | F6 | - |
| F2 | Track Record/Past Performance | 6 | F3 | 0.15 |
| F3 | Technical Ability | 2 | F7 | 0.15 |
| F4 | Quality of Staff | 7 | F5 | 0.05 |
| F5 | Risk Management and Resilience | 4 | F9 | 0.10 |
| F6 | Ability to Fulfill on Schedule | 1 | F2 | 0.15 |
| F7 | Financial Capability | 3 | F4 | 0.10 |
| F8 | Market Viability | 10 | F13 | 0.15 |
| F9 | Management Ability | 5 | F10 | 0.10 |
| F10 | Pricing and Cost | 9 | F8 | 0.05 |
| F11 | Information Sharing | 11 | F11 | 0.05 |
| F12 | Research and Innovation | 12 | F12 | 0.05 |
| F13 | Availability and Performance of Ships and Equipment | 8 | F1 | 0.05 |

**Table 19.** Function Calculation of Relative Importance to Expert 1.

| New Ranking | S$j$ | $k_j$ |
|---|---|---|
| F6 Ability to Fulfill on Schedule | – | 1 |
| F3 Technical Ability | 0.15 | 1.15 |
| F7 Financial Capability | 0.15 | 1.15 |
| F5 Risk Management and Resilience | 0.05 | 1.05 |
| F9 Management Ability | 0.10 | 1.10 |
| F2 Track Record | 0.15 | 1.15 |
| F4 Quality of Staff | 0.10 | 1.10 |
| F13 Availability and Performance of Ships and Equipment | 0.15 | 1.15 |
| F10 Pricing and Cost | 0.10 | 1.10 |
| F8 Market Viability | 0.05 | 1.05 |
| F11 Information Sharing | 0.05 | 1.05 |
| F12 Research and Innovation | 0.05 | 1.05 |
| F1 Company Reputation | 0.05 | 1.05 |

Step 3.   Calculate the initial weight $q_j$ according to Formula (13), as shown in Table 20.

**Table 20.** Initial weight calculation of Expert 1.

| New Ranking | S$j$ | $k_j$ | $q_j$ |
|---|---|---|---|
| F6 Ability to Fulfill on Schedule | - | 1 | 1 |
| F3 Technical Ability | 0.15 | 1.15 | 0.870 |
| F7 Financial Capability | 0.15 | 1.15 | 0.757 |
| F5 Risk Management and Resilience | 0.05 | 1.05 | 0.721 |
| F9 Management Ability | 0.10 | 1.10 | 0.655 |
| F2 Track Record | 0.15 | 1.15 | 0.570 |
| F4 Quality of Staff | 0.10 | 1.10 | 0.518 |
| F13 Availability and Performance of Ships and Equipment | 0.15 | 1.15 | 0.450 |
| F10 Pricing and Cost | 0.10 | 1.10 | 0.410 |
| F8 Market Viability | 0.05 | 1.05 | 0.390 |
| F11 Information Sharing | 0.05 | 1.05 | 0.371 |
| F12 Research and Innovation | 0.05 | 1.05 | 0.353 |
| F1 Company Reputation | 0.05 | 1.05 | 0.336 |

Step 4.   Obtain the final weight $w_j$ according to Formula (14), as shown in Table 21.

**Table 21.** Final weight calculation of Expert 1.

| New Ranking | $S_j$ | $k_j$ | $q_j$ | $w_j$ |
|---|---|---|---|---|
| F6 Ability to Fulfill on Schedule | - | 1 | 1 | 0.135 |
| F3 Technical Ability | 0.15 | 1.15 | 0.870 | 0.118 |
| F7 Financial Capability | 0.15 | 1.15 | 0.757 | 0.102 |
| F5 Risk Management and Resilience | 0.05 | 1.05 | 0.721 | 0.098 |
| F9 Management Ability | 0.10 | 1.10 | 0.655 | 0.089 |
| F2 Track Record | 0.15 | 1.15 | 0.570 | 0.077 |
| F4 Quality of Staff | 0.10 | 1.10 | 0.518 | 0.070 |
| F13 Availability and Performance of Ships and Equipment | 0.15 | 1.15 | 0.450 | 0.061 |
| F10 Pricing and Cost | 0.10 | 1.10 | 0.410 | 0.054 |
| F8 Market Viability | 0.05 | 1.05 | 0.390 | 0.053 |
| F11 Information Sharing | 0.05 | 1.05 | 0.371 | 0.050 |
| F12 Research and Innovation | 0.05 | 1.05 | 0.353 | 0.048 |
| F1 Company Reputation | 0.05 | 1.05 | 0.336 | 0.045 |
| Subtotal | | | 7.401 | 1 |

Step 5. Calculate the arithmetic average of the weights from all the experts:

According to Step 2 to Step 4 of this section, calculate the weight values from all the experts and obtain the average values. Prioritize the values again, the greater the value, the higher the ranking, as shown in Table 22.

**Table 22.** The arithmetic averages of the weights from all the experts.

| Item | Expert 1 Weights | Expert 2 Weights | Expert 3 Weights | Expert 4 Weights | Average Weights | New Ranking |
|---|---|---|---|---|---|---|
| F1 | 0.045 | 0.103 | 0.070 | 0.081 | 0.075 | 6 |
| F2 | 0.077 | 0.119 | 0.122 | 0.103 | 0.105 | 2 |
| F3 | 0.118 | 0.090 | 0.140 | 0.130 | 0.119 | 1 |
| F4 | 0.070 | 0.081 | 0.052 | 0.061 | 0.066 | 10 |
| F5 | 0.098 | 0.067 | 0.063 | 0.070 | 0.074 | 7 |
| F6 | 0.135 | 0.078 | 0.084 | 0.113 | 0.103 | 3 |
| F7 | 0.102 | 0.058 | 0.055 | 0.074 | 0.072 | 8 |
| F8 | 0.053 | 0.055 | 0.047 | 0.055 | 0.053 | 11 |
| F9 | 0.089 | 0.061 | 0.076 | 0.089 | 0.079 | 5 |
| F10 | 0.054 | 0.074 | 0.096 | 0.048 | 0.068 | 9 |
| F11 | 0.050 | 0.048 | 0.043 | 0.040 | 0.045 | 13 |
| F12 | 0.048 | 0.053 | 0.041 | 0.042 | 0.046 | 12 |
| F13 | 0.061 | 0.113 | 0.111 | 0.094 | 0.095 | 4 |

Based on the results, the importance ranking of the influence factors of partnership selection is F3 > F2 > F6 > F13 > F9 > F1 > F5 > F7 > F10 > F4 > F8 > F12 > F11.

4.3.2. Selection of the Most Suitable Partnership

Three experts (the same as those who marked the scores in Section 4.2.2) were invited to refer to Table 8 for background information of the partners and then scored according to the definition in Section 3.2.2.

Step 1.    Expert Evaluation:

Experts were asked to subjectively use the linguistic variables listed in Table 7 and evaluate the influence factor F of each Plan P; the table of linguistic variable evaluation was then converted into triangular fuzzy numbers, as shown in Table 23.

Step 2.    Establish fuzzy decision matrix:

**Table 23.** Expert evaluation of triangular fuzzy numbers.

| Influence Factor | Expert 1 | | | Expert 2 | | | Expert 3 | | |
|---|---|---|---|---|---|---|---|---|---|
| | **P1** | **P2** | **P3** | **P1** | **P2** | **P3** | **P1** | **P2** | **P3** |
| F1 | 7, 9, 10 | 5, 7, 9 | 5, 7, 9 | 9, 10, 10 | 5, 7, 9 | 7, 9, 10 | 9, 10, 10 | 7, 9, 10 | 9, 10, 10 |
| F2 | 9, 10, 10 | 7, 9, 10 | 9, 10, 10 | 7, 9, 10 | 3, 5, 7 | 7, 9, 10 | 9, 10, 10 | 5, 7, 9 | 9, 10, 10 |
| F3 | 7, 9, 10 | 7, 9, 10 | 5, 7, 9 | 7, 9, 10 | 5, 7, 9 | 5, 7, 9 | 7, 9, 10 | 7, 9, 10 | 7, 9, 10 |
| F4 | 7, 9, 10 | 3, 5, 7 | 5, 7, 9 | 7, 9, 10 | 7, 9, 10 | 7, 9, 10 | 7, 9, 10 | 3, 5, 7 | 5, 7, 9 |
| F5 | 7, 9, 10 | 7, 9, 10 | 3, 5, 7 | 3, 5, 7 | 5, 7, 9 | 7, 9, 10 | 5, 7, 9 | 3, 5, 7 | 7, 9, 10 |
| F6 | 7, 9, 10 | 7, 9, 10 | 7, 9, 10 | 7, 9, 10 | 3, 5, 7 | 5, 7, 9 | 9, 10, 10 | 5, 7, 9 | 5, 7, 9 |
| F7 | 9, 10, 10 | 3, 5, 7 | 3, 5, 7 | 7, 9, 10 | 3, 5, 7 | 3, 5, 7 | 7, 9, 10 | 5, 7, 9 | 5, 7, 9 |
| F8 | 7, 9, 10 | 1, 3, 5 | 1, 3, 5 | 5, 7, 9 | 5, 7, 9 | 5, 7, 9 | 7, 9, 10 | 3, 5, 7 | 9, 10, 10 |
| F9 | 7, 9, 10 | 3, 5, 7 | 1, 3, 5 | 7, 9, 10 | 3, 5, 7 | 5, 7, 9 | 5, 7, 9 | 7, 9, 10 | 5, 7, 9 |
| F10 | 3, 5, 7 | 5, 7, 9 | 5, 7, 9 | 3, 5, 7 | 7, 9, 10 | 3, 5, 7 | 3, 5, 7 | 3, 5, 7 | 7, 9, 10 |
| F11 | 9, 10, 10 | 3, 5, 7 | 5, 7, 9 | 9, 10, 10 | 5, 7, 9 | 5, 7, 9 | 7, 9, 10 | 7, 9, 10 | 9, 10, 10 |
| F12 | 7, 9, 10 | 5, 7, 9 | 5, 7, 9 | 9, 10, 10 | 7, 9, 10 | 9, 10, 10 | 7, 9, 10 | 3, 5, 7 | 9, 10, 10 |
| F13 | 7, 9, 10 | 5, 7, 9 | 3, 5, 7 | 9, 10, 10 | 5, 7, 9 | 7, 9, 10 | 9, 10, 10 | 3, 5, 7 | 7, 9, 10 |

Construct the fuzzy decision matrix as shown in Table 24 according to Formula (16).

Step 3.    Establish a normalized fuzzy decision matrix:

**Table 24.** Fuzzy decision matrix.

| Plan / Factor | Partnership P1 | Partnership P2 | Partnership P3 |
|---|---|---|---|
| F1 | 7, 9.667, 10 | 5, 7.667, 10 | 5, 8.667, 10 |
| F2 | 7, 9.667, 10 | 3, 7, 10 | 7, 9.667, 10 |
| F3 | 7, 9, 10 | 5, 8.333, 10 | 5, 7.667, 10 |
| F4 | 7, 9, 10 | 3, 6.333, 10 | 5, 7.667, 10 |
| F5 | 3, 7, 10 | 3, 7, 10 | 3, 7.667, 10 |
| F6 | 7, 9.333, 10 | 3, 7, 10 | 5, 7.667, 10 |
| F7 | 7, 9.333, 10 | 3, 5.667, 9 | 3, 5.667, 9 |
| F8 | 5, 8.333, 10 | 1, 5, 9 | 1, 6.667, 10 |
| F9 | 5, 8.333, 10 | 3, 6.333, 10 | 1, 5.667, 9 |
| F10 | 3, 5, 7 | 3, 7, 10 | 3, 7, 10 |
| F11 | 7, 9.667, 10 | 3, 7, 10 | 5, 8, 10 |
| F12 | 7, 9.333, 10 | 3, 7, 10 | 5, 9, 10 |
| F13 | 7, 9.667, 10 | 3, 6.333, 9 | 3, 7.667, 10 |

Use the fuzzy decision matrix to calculate and obtain the normalized fuzzy number with Step 3, Equation (17) in Section 3.2.2, as shown in Table 25.

Step 4.    Establish a weighted normalized fuzzy decision matrix:

Multiply the SWARA weight obtained in Section 4.3.1 by the matrix in Table 25 to obtain the weighted normalized fuzzy decision matrix, as shown in Table 26.

**Table 25.** Normalized fuzzy decision matrix.

| Factor | Plan | Partnership P1 | Partnership P2 | Partnership P3 |
|---|---|---|---|---|
| F1 | | 0.7, 0.967, 1 | 0.5, 0.767, 1 | 0.5, 0.867, 1 |
| F2 | | 0.7, 0.967, 1 | 0.3, 0.7, 1 | 0.7, 0.967, 1 |
| F3 | | 0.7, 0.9, 1 | 0.5, 0.833, 1 | 0.5, 0.767, 1 |
| F4 | | 0.7, 0.9, 1 | 0.3, 0.633, 1 | 0.5, 0.767, 1 |
| F5 | | 0.3, 0.7, 1 | 0.3, 0.7, 1 | 0.3, 0.767, 1 |
| F6 | | 0.7, 0.933, 1 | 0.3, 0.7, 1 | 0.5, 0.767, 1 |
| F7 | | 0.7, 0.933, 1 | 0.3, 0.567, 0.9 | 0.3, 0.567, 0.9 |
| F8 | | 0.5, 0.833, 1 | 0.1, 0.5, 0.9 | 0.1, 0.667, 1 |
| F9 | | 0.5, 0.833, 1 | 0.3, 0.633, 1 | 0.1, 0.567, 0.9 |
| F10 | | 0.429, 0.6, 1 | 0.333, 0.45, 1 | 0.333, 0.45, 1 |
| F11 | | 0.7, 0.967, 1 | 0.3, 0.7, 1 | 0.5, 0.8, 1 |
| F12 | | 0.7, 0.933, 1 | 0.3, 0.7, 1 | 0.5, 0.9, 1 |
| F13 | | 0.7, 0.967, 1 | 0.3, 0.633, 0.9 | 0.3, 0.767, 1 |

Step 5.   Calculate the fuzzy positive and negative ideal solution vectors:

**Table 26.** Weighted normalized fuzzy decision matrix.

| Influence Factor | SWARA Weight | Partnership P1 | Partnership P2 | Partnership P3 |
|---|---|---|---|---|
| F1 | 0.075 | 0.053, 0.073, 0.075 | 0.038, 0.058, 0.075 | 0.038, 0.065, 0.075 |
| F2 | 0.105 | 0.074, 0.102, 0.105 | 0.032, 0.074, 0.105 | 0.074, 0.102, 0.105 |
| F3 | 0.119 | 0.083, 0.107, 0.119 | 0.060, 0.099, 0.119 | 0.060, 0.091, 0.119 |
| F4 | 0.066 | 0.046, 0.059, 0.066 | 0.020, 0.042, 0.066 | 0.033, 0.051, 0.066 |
| F5 | 0.074 | 0.022, 0.052, 0.074 | 0.022, 0.052, 0.074 | 0.022, 0.057, 0.074 |
| F6 | 0.103 | 0.072, 0.096, 0.103 | 0.031, 0.072, 0.103 | 0.052, 0.079, 0.103 |
| F7 | 0.072 | 0.050, 0.067, 0.072 | 0.022, 0.041, 0.065 | 0.022, 0.041, 0.065 |
| F8 | 0.053 | 0.027, 0.044, 0.053 | 0.005, 0.027, 0.048 | 0.005, 0.035, 0.053 |
| F9 | 0.079 | 0.040, 0.066, 0.079 | 0.024, 0.050, 0.079 | 0.008, 0.045, 0.071 |
| F10 | 0.068 | 0.029, 0.041, 0.068 | 0.023, 0.031, 0.068 | 0.023, 0.031, 0.068 |
| F11 | 0.045 | 0.032, 0.044, 0.045 | 0.014, 0.032, 0.045 | 0.023, 0.036, 0.045 |
| F12 | 0.046 | 0.032, 0.043, 0.046 | 0.014, 0.032, 0.046 | 0.023, 0.041, 0.046 |
| F13 | 0.095 | 0.067, 0.092, 0.095 | 0.029, 0.060, 0.086 | 0.029, 0.073, 0.095 |

Calculate the fuzzy positive ideal solution $V^+$ and the fuzzy negative ideal solution $V^-$ using Equations (19) and (20), as shown in Table 27.

Step 6.   Calculating the positive and negative degrees of separation:

**Table 27.** Fuzzy positive and negative ideal solution vectors.

| Vector Factor | $V^+$ | $V^-$ |
|---|---|---|
| F1 | 0.053, 0.073, 0.075 | 0.038, 0.058, 0.075 |
| F2 | 0.074, 0.102, 0.105 | 0.032, 0.074, 0.105 |
| F3 | 0.083, 0.107, 0.119 | 0.060, 0.091, 0.119 |
| F4 | 0.046, 0.059, 0.066 | 0.020, 0.042, 0.066 |
| F5 | 0.022, 0.057, 0.074 | 0.022, 0.052, 0.074 |
| F6 | 0.072, 0.096, 0.103 | 0.031, 0.072, 0.103 |
| F7 | 0.050, 0.067, 0.072 | 0.022, 0.041, 0.065 |
| F8 | 0.027, 0.044, 0.053 | 0.005, 0.027, 0.048 |
| F9 | 0.040, 0.066, 0.079 | 0.008, 0.045, 0.071 |
| F10 | 0.029, 0.041, 0.068 | 0.023, 0.031, 0.068 |
| F11 | 0.032, 0.044, 0.045 | 0.014, 0.032, 0.045 |
| F12 | 0.032, 0.043, 0.046 | 0.014, 0.032, 0.046 |
| F13 | 0.067, 0.092, 0.095 | 0.029, 0.060, 0.086 |

Use Equation (21) to obtain the positive degree of separation $d_i^+$, as shown in Table 28, and use Equation (22) to obtain the negative degree of separation $d_i^-$, as shown in Table 29.

Step 7. Calculate and sort the coefficients of determination:

**Table 28.** Positive degree of separation.

| Influence Factor | Partnership P1 | Partnership P2 | Partnership P3 |
|---|---|---|---|
| F1 | 0 | 0.01225 | 0.00981 |
| F2 | 0 | 0.02914 | 0 |
| F3 | 0 | 0.01407 | 0.00924 |
| F4 | 0 | 0.01794 | 0.00883 |
| F5 | 0.00289 | 0.00289 | 0 |
| F6 | 0 | 0.02742 | 0.01517 |
| F7 | 0 | 0.02207 | 0.02207 |
| F8 | 0 | 0.01631 | 0.01371 |
| F9 | 0 | 0.01308 | 0.02665 |
| F10 | 0 | 0.00671 | 0.00671 |
| F11 | 0 | 0.01249 | 0.00458 |
| F12 | 0 | 0.01217 | 0.00529 |
| F13 | 0 | 0.02915 | 0.02454 |
| $d_i^+$ | 0.00289 | 0.21569 | 0.14660 |

**Table 29.** Negative degree of separation.

| Influence Factor | Partnership P1 | Partnership P2 | Partnership P3 |
|---|---|---|---|
| F1 | 0.01225 | 0 | 0.00404 |
| F2 | 0.02914 | 0 | 0.02914 |
| F3 | 0.01619 | 0.00462 | 0 |
| F4 | 0.01794 | 0 | 0.03271 |
| F5 | 0 | 0 | 0.00289 |
| F6 | 0.002742 | 0 | 0.01277 |
| F7 | 0.02207 | 0 | 0 |
| F8 | 0.01631 | 0 | 0.00548 |
| F9 | 0.02665 | 0.01072 | 0 |

**Table 29.** *Cont.*

| Influence Factor | Partnership P1 | Partnership P2 | Partnership P3 |
|---|---|---|---|
| F10 | 0.00671 | 0 | 0 |
| F11 | 0.01249 | 0 | 0.00566 |
| F12 | 0.01217 | 0 | 0.00735 |
| F13 | 0.02915 | 0 | 0.00911 |
| $(d_i^-)$ | 0.22849 | 0.01534 | 0.10915 |

Use Equation (23) to calculate the *CCi* value of each candidate in order to determine the ranking order of all partners. The one with the highest *CCi* value will be the best partner choice, as shown in Table 30.

**Table 30.** Determining coefficients and rankings.

| Plan<br>Factor | Partnership P1 | Partnership P2 | Partnership P3 |
|---|---|---|---|
| Positive degree of separation $(d_i^+)$ | 0.00289 | 0.21569 | 0.14660 |
| Negative degree of separation $(d_i^-)$ | 0.22849 | 0.01534 | 0.10915 |
| Coefficient of determination $(CC_i)$ | 0.98751 | 0.06640 | 0.42678 |
| Ranking | 1 | 3 | 2 |

*4.4. Determining the Best Decision Plan*

The overall investment risk value obtained in this case is within the permitted range, which means that it is viable for the underwater foundations manufacturers to invest in the offshore wind power industry. The decision-making model shows that P1 is the best partnership choice, followed by P3, and P2 ranks the third.

This article aims to provide some valuable enlightenment for future researchers to promote the development of offshore wind power research. It is also essential to apply interdisciplinary methods and establish appropriate evaluation and optimization models. Establishing a comprehensive decision-making framework or model for multi-dimensional evaluation, and then transforming it into multiple sub-standards for analysis, is considered to be future work. These methods and models can not only describe uncertainty, but also consider flexibility and strategies in the process, Future research will be to construct a model from a micro perspective and extend it to macro analysis in certain scenarios.

**5. Conclusions and Suggestions**

*5.1. Conclusions*

Based on the research process and results presented in the four chapters above, four conclusions can be summarized as follows:

1.  Previous research on evaluation models mostly explored the choice of a single-frame evaluation model or a single-perspective evaluation structure, which could cause deviations in the evaluation results. This study, on the other hand, establishes a different composite evaluation model and successfully integrates several evaluation methods so as to obtain a more objective evaluation result.
2.  The study applied FPR to calculate the weight of the risk factors, along with the scores, to determine investment risk in the offshore wind power foundations industry. In addition, SWARA was adopted to calculate the weight of partners' criteria, followed by FTOPSIS to select the most suitable partner. The study serves as a valuable reference for the firms in their decision-making process.
3.  The "Investment Evaluation and Partner Selection Model in the Offshore Wind Power Underwater Foundations Industry" established in this study simplifies what was formerly a lengthy and complicated selection process. Throughout the evaluation

process, only decision makers' intuitive judgement data are required in order to quickly and accurately generate fair and accurate evaluations and recommendations.

4. Using the criteria and weights analyzed in this research as a guide, the firms in the supply chain can proactively raise their professional thresholds, whereby they can accelerate the development of offshore wind power and its related industries. Once training and professional skills are acquired, local wind farms in Taiwan can then expand into the Asia-Pacific market going forward.

*5.2. Suggestions*

Based on the results of this research, the following three suggestions are summarized:

1. This research only focuses on the underwater basic industries that currently have advantages in offshore wind power localization. Follow-up research can be expanded to other localization projects such as wind turbine components, submarine cables, offshore work ships, and maritime engineering.
2. The criteria for the compilation of this study and the scale of potential partners are only discussed based on existing data. After the development of Taiwanese wind farms in the future, more and more updated data will be collected. Follow-up research can be combined with artificial intelligence methods to refine the evaluation model. Besides, the proposed framework relies on risk preference parameters. Since risk preferences vary from person to person, introducing machine learning methods such as neural network and genetic programming to simulate decision preferences can not only improve evaluation efficiency, but also ensure model accuracy, which will be an important future research agenda.
3. Due to the small number of wind farms developed and constructed in Taiwan, this study can only use a single case for model verification. It is recommended that future studies explore more cases for model verification and comparison.

**Author Contributions:** M.-Y.C. conceived of the main research idea, provided extensive advice throughout the study, and made methodological revisions; Y.-F.W. collected the data, conducted the expert interviews, performed calculations, and analyzed the data and results; Y.-F.W. administrated the project and wrote the paper; and M.-Y.C. and Y.-F.W. discussed the model evaluation results and commented on the paper. All authors have read and agreed to the published version of the manuscript.

**Funding:** This research received no external funding.

**Institutional Review Board Statement:** Not applicable.

**Informed Consent Statement:** Not applicable.

**Data Availability Statement:** Not applicable.

**Conflicts of Interest:** The authors declare no conflict of interest.

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
