# Peer review of "Investment Evaluation and Partnership Selection Model in the Offshore Wind Power Underwater Foundations Industry"

_jmse, doi:10.3390/jmse9121371_

Round 1
Reviewer 1 Report
Investment Evaluation and Partnership Selection Model in the 2 Offshore Wind Power Underwater Foundations Industry
The paper presents a decision model for the partnership of companies in the underwater foundation of the wind power industry. The multi-criteria decision-making model has been constructed for the problem. Although the problem of the paper is interesting, there are some key issues that need to be addressed:
Major revisions:
- I think in the abstract the authors should explain why this paper is unique or novel theoretically and practically.
- The contributions of the paper should be also discussed in the literature review section (section 2).
- To evaluate the developed methodology, the results of SWARA-FTOPSIS methods should also be compared with other similar methods. For example, you can see the following studies: “ Risk-based material selection process supported on information theory: a case study on industrial gas turbine. Applied Soft Computing, (2017)52, 1116-1129.” and “Comprehensive MULTIMOORA method with target-based attributes and integrated significant coefficients for materials selection in biomedical applications. Materials & Design, (2015) 87, 949-959.”
- I think some of the results and sensitivity analysis can be better illustrated by figures.
- I think it is better to mention the main limitations and future research directions in this section.
- Please add some discussions for future researches in the conclusion section.
- The paper has an appropriate flow and acceptable English, language readability, and fluency; however, I think it could be strengthened for various readers of the paper.
Reviewer 2 Report
The paper is well-written and provides interesting research. The test method is obvious and justified, the research data are appropriate and adequate, and the authors accurately reported the study. It is critical that the test method be repeatable by other scientists. The following comprehensive remarks are intended to address some of the article's flaws.
I would suggest include the following reference in the introduction for additional reasoning.
https://doi.org/10.1016/j.egyr.2020.09.035, https://doi.org/10.3390/su13169060 https://doi.org/10.3390/su13084524,
All the best and stay safe
Reviewer 3 Report
The present article is of a good scientific level, but the following points are recommended to improve the work:
1. There are many spelling and grammar flaws.
2. The results of the present work should be stated in the abstract.
3. For all equations, reference is required.
4. Avoid batch referencing, such as [8-17].
